# Relevance Analysis of Sustainable Development of China’s Yangtze River Economic Belt Based on Spatial Structure

**DOI:** 10.3390/ijerph16173076

**Published:** 2019-08-23

**Authors:** Decai Tang, Zhijiang Li, Brandon J. Bethel

**Affiliations:** 1China Institute of Manufacturing Development, Nanjing University of Information Science & Technology, Nanjing 210044, China; 2School of Management Science and Engineering, Nanjing University of Information Science & Technology, Nanjing 210044, China; 3School of Marine Sciences, Nanjing University of Information Science & Technology, Nanjing 210044, China

**Keywords:** China’s Yangtze River Economic Belt, space structure, sustainable development, relevance analysis

## Abstract

Scientifically justifiable spatial structure can not only promote the efficient use of regional resources, but can also effectively avoid “urban diseases”, such as traffic congestion, housing shortage, resource scarcity, and so on. It is the “regulator” and “booster” of regional development. Firstly, this paper measures the spatial structure of the Yangtze River Economic Belt from the four dimensions of scale distribution, central structure, spatial connection, and compactness: Gini coefficient of urban scale, urban primacy, regional economic linkage strength, and spatial compactness. Secondly, the optimized Super-Slack Based Measure-Undesirable model is used to evaluate the sustainable development status of the Yangtze River Economic Belt. Finally, a sustainable development correlation analysis model based on regional spatial structure is constructed. Based on the overall perspective of the Yangtze River Economic Belt and the individual perspective of 11 provinces and cities, the relationship between the spatial structure of the Yangtze River Economic Belt and sustainable development is analyzed. It is found that the impact of the four spatial structure indicators on the sustainable development level of the Yangtze River Economic Zone is relatively stable in five different periods. The ranking results are as follows: Gini coefficient of urban scale > urban primacy > regional economic linkage strength > spatial compactness.

## 1. Introduction

The Yangtze River Economic Zone covers 11 provinces and cities (Guizhou, Sichuan, Yunnan, Chongqing, Hunan, Hubei, Jiangxi, Anhui, Jiangsu, Zhejiang and Shanghai), accounts for more than 40% of China’s population and total economic output, and occupies 21% of China’s land area [1]. It is the center and backbone of China’s economic and cultural territory. From the initial proposal of the “one line, one axis” strategy in 1980, General Secretary Xi Jinping in 2018 held a symposium promoting the development of the Yangtze River Economic Belt (YREB) in which the acceleration of sustainable development of the YREB was promoted by building a new supporting belt for economic growth.

In recent years, the advantages of rapid development previously achieved by low costs of labor as induced by the large population, resource consumption, and environmental degradation are gradually disappearing. However, traditional socioeconomic development has not yet fundamentally changed, which has resulted in the slowdown of economic growth in the YREB, leading to a developmental bottleneck. Simultaneously, problems such as resource shortages, environmental pollution, ecological destruction, and excess capacity of traditional industries are emerging, which ultimately threaten the sustainable development of all provinces and municipalities and even the YREB [2]. On the one hand, with the rapid progress of urbanization, regional spatial structure and its socioeconomic performance has attracted the wide attention of academia. Spatial structure adapted to local conditions is the “regulator” and “booster” required to promote regional high-quality development [3,4]. Presently, scientific research on regional spatial structure and its evolutions is still at the initial stage with an accompanying dearth of the empirical analyses on the characteristics of regional spatial structure and socio-economic performance. For certain regions, if the influence of regional spatial structure is not fully considered, the formulation of regional planning will inevitably lead to the loss of efficiency of the whole region and individual cities within the region. Therefore, studies based on regional spatial structure will become an important part of regional sustainable development in the future. The YREB spans China’s eastern, western and central regions (the western reaches are Guizhou, Sichuan, Yunnan, and Chongqing; the central reaches are Hunan, Hubei, Jiangxi, and Anhui; the eastern reaches are Jiangsu, Zhejiang, and Shanghai). There are significant discrepancies in both resource endowment and socioeconomic development across the 11 provinces and cities. Moreover, the integration pattern is far from being formed, and the development trace of internal fragmentation and non-equilibrium is still very obvious [5]. Therefore, quantitative analysis of the relationship between spatial structure and sustainable development of the 11 provinces and cities in the YREB from the perspective of spatial structure, is a key step to accelerate the sustainable development of the YREB. It provides a reference for further study of the interaction between human social development activities and environmental evolution law and seeking the way of human social and environmental co-evolution.

## 2. Research Status

Regional spatial structure is an important factor affecting the input–output efficiency of various factors of production in the region and is the key to sustainable development. A spatial structure organized in a scientific and rational way can not only optimize the allocation of various factors of production and achieve the best use of goods, but also achieve complementary advantages through the construction of inter-city networks, resulting in a multiplicity of outputs [6]. On the one hand, upon summarizing the existing research results, one finds that the policy research on spatial structure and regional development are numerous and mainly focuses on answering the following three questions: Firstly, which spatial structure model can promote regional sustainable development better in expansive cities and compact cities [7]? Secondly, will the excessively close social and economic links among the cities in the region affect the sustainable development of the cities themselves [8]? Thirdly, will a larger city scale be a stumbling block to the sustainable development of cities [9]?

On the other hand, research results based on empirical analysis are relatively few. Scholars such as Fallah [10], Lee and Gordon [11], Meijers and Burger [12] conducted empirical analyses on the relationship between regional spatial structure and socio-economic development in the United States. Firstly, Fallah et al. studied the relationship between the regional expansion intensity and regional productivity level in the United States, and found that there was a significant negative correlation between them [10]. Secondly, Lee and Gordon quantitatively expressed regional development activities through employment growth, and then studied the role of regional spatial structure in regional development. The results show that the impact of regional spatial structure on regional economic development depends on the size of the city, while the smaller scale shows faster economic growth, and vice versa [11]. Finally, based on Lee and Gordon’s research, Meijers and Burger found that areas with a higher central structure had higher productivity [12]. Zhang Peng, a Chinese scholar, takes provincial administrative units as the research object, and found that the single-center spatial structure of provincial regions in China has a certain role in promoting the regional socio-economic effect [13]. Zhong Yexi et al. found that spatial connection is the most important factor affecting regional sustainable development based on the comparative study of distance, spatial connection, and scale among cities in the region [14].

The research on the relationship between spatial structure and regional development has the following characteristics and trends: First, when discussing which spatial structure can promote sustainable regional socio-economic development, results are very different. Secondly, there are relatively few quantitative studies on the relationship between spatial structure and sustainable development. Thirdly, the existing research on spatial structure and regional development is based on the perspective of a certain spatial structure. Few studies on the relationship between spatial structure and regional development are based on a variety of spatial structures and carry out comparative analyses.

## 3. Indicators Selection, Econometric Model, and Number Illustration

### 3.1. Indicators Selection

Summarizing the existing research results on the impact of spatial structure on regional socio-economic development, this paper chooses four regional spatial structure indicators, namely, the Gini coefficient of urban scale, urban primacy, regional economic linkage strength, and spatial compactness, from four dimensions of scale distribution, central structure, spatial connection, and compactness to characterize the spatial structure of the 11 provinces and cities of the YREB. Thus, it can reflect not only the spatial distribution and combination of cities in the region, but also the relationship network between cities [15].

#### (1) Dimension of Scale Distribution

Based on the existing research results and the development status of the 11 provinces and cities in the YREB, the Gini coefficient of urban scale is selected to measure whether a reasonable urban hierarchy has been formed within the region. The Gini coefficient is first used to reflect the gap between rich and poor in a country or region. Marshall applied the Gini coefficient to regional development to reflect disparities between cities within a region and so it is also called the Gini coefficient of urban scale [16]. The Gini coefficient of urban scale ranges from 0 to 1. If the value approaches 0, it shows that there is a small gap between cities and the distribution of various development factors is more balanced. If the trend is closer to 1, it shows that there is a big gap in the scale of cities in the region, and all kinds of development factors are clustered in a few cities [17]. The Gini coefficient of urban scale (*G*) is calculated as follows:(1)$$G=\frac{T}{2S\left(n-1\right)}=\frac{\sum_{j=1}^{n}\sum_{i=1}^{n}\|x_{i}-x_{j}\|}{2S\left(n-1\right)}$$
where *S* represents the total GDP of the whole region, *n* represents the number of cities, and *T* represents the absolute sum of the difference of GDP scale between each city in the region [18].

#### (2) Central Structure

According to the growth pole theory put forward by Perroux, regional economic growth can spread outward from nodes, and then drive regional development. Therefore, the radiation effect of regional central cities on surrounding cities and regions is also the key research content of regional spatial structure. Based on central structure, this paper employs urban primacy to measure the radiation and influence of the central city in the region. Jefferson defined urban primacy as the ratio of the population size of the first city to that of the second city [19]. With the deepening of the research on urban spatial structure, an increasing number of researchers use this index to measure the degree of aggregation of regional comprehensive development factors in central cities, and to study the radiation and influence of central cities in a given region. Many research results show that the higher the urban primacy is, the more inclined the regional development factors are to the first city. However, if the city is overprioritized, it will not only fail to bring about regional development benefits, but also will hinder its own development, resulting in various urban diseases. The two-city index method is a common method to calculate the primacy degree, but some scholars believe that although the two-city index is easy to understand and calculate, it is inevitably partial. Zhou proposed the four-city index method and the eleven-city index method to improve the two-city index [20]. They are calculated are as follows:(2)$$\begin{array}{c} S_{1}=P_{1}/P_{2}\\ S_{3}=P_{1}/\left(P_{2}+P_{3}+\ldots +P_{11}\right)\\ S_{2}=P_{1}/\left(P_{2}+P_{3}+P_{4}\right) \end{array}$$
where *S*_1_, *S*_2_, and *S*_3_ represent the urban primacy calculated by the two-city index method, the four-city index method and the eleven-city index method respectively. *P*_1_, *P*_2_, and *P*_3_...*P*_11_ represent the population size of the first city to the eleventh city respectively. Because in many practical applications, the four-city index method and the eleven-city index method do not show obvious advantages, so this paper still uses the two-city index method to calculate the urban primacy of 11 provinces and cities in the YREB.

#### (3) Spatial Connection Dimension

According to the theory of spatial interaction, cities in the region are not static and closed, but dynamic and open. There are intricate connection networks among cities in a given region through convection, radiation, and transmission. The interconnection between cities in the region is the concrete manifestation of the flow of various factors of production in the region, and also the internal motive force for the formation and reorganization of regional spatial structure. Among many regional spatial linkages, the importance of economic linkages is most prominent. Therefore, considering the complexity of regional spatial linkages and the importance of economic linkages, this paper refers to relevant research results to represent regional spatial linkages by regional economic linkages [21]. The measurement of regional economic linkages is becoming more and more mature which include but are not limited to the including gravitational [22], fracture point measurement [23], and Wilson measurement [24] models. This paper uses the gravitational model to measure the regional economic linkages of 11 provinces and cities in the YREB. The equation is as follows:(3)$$r_{ij}=\frac{\sqrt{P_{i}\times G_{i}}\times \sqrt{P_{j}\times G_{j}}}{D_{ij}^{2}}$$
where *r_ij_* denotes the regional economic linkage strength between two cities *P_i_* and *P_j_* denote the number of urban employment and *G_i_* and *G_j_* denote the GDP, and *D_ij_* denotes the straight-line distance between the two cities of a given region.

Incorporating lessons learned from other related studies, this study improves the above model following two crucial aspects [25]. Firstly, the ratio of urban GDP to the sum of urban GDP of the two cities is the coefficient of increase. Secondly, in view of the transportation status of 11 provinces and cities in the YREB, this paper takes the shortest road distance between the two cities as the calculation basis. The revised formula is as follows:(4)$$R_{ij}=\frac{\sum_{i,j=1}^{n}k_{ij}\frac{\sqrt{P_{i}\times G_{i}}\times \sqrt{P_{j}\times G_{j}}}{D_{ij}^{2}}}{1+2+\cdots +\left(n-1\right)},\;\mathrm{and}\;k_{ij}=\frac{G_{i}}{G_{i}+G_{j}}$$
where *R_ij_* is the average regional economic linkage strength between two cities, *P_i_* and *P_j_* is the number of the employed within a given region, *G_i_* and *G_j_* represent the GDP between two cities, *D_ij_* is the highway distance between the two cities and *k_ij_* is the contribution rate of city *i* to *r_ij_*.

#### (4) Regional Spatial Compactness

Spatial compactness refers to the degree of spatial concentration of developmental factors such as towns, enterprises, factories, funds, facilities, technology, and expertise in the process of regional development and is the basis of promoting regional intensive and efficient development. Good spatial compactness is the concentrated expression of maximizing regional comprehensive benefits. It can promote the efficient utilization of resources, high spatial output and high-density development. However, too high or too low spatial compactness is not conducive to healthy regional development which can lead to traffic congestion, an excessive concentration of industry and population, and other negative effects. Compact cities are a form of city that promotes the sustainable development of a city from the perspective of the city itself. It is one of the representative theories to promote the sustainable development of cities, which is based on the premise of curbing urban expansion, and through the centralized setting of public facilities, reducing traffic distance, and reducing pollution emissions. Similarly, compact regions promote sustainable development by promoting sustainable development in multiple cities. Compact regions are more resource-saving, high-density and high-efficiency urban architecture as compared to more decentralized areas. As the name suggests, the compactness of regional space index is used to, of course, measure the compactness of regional space. Here, it is expressed as a ratio of the total amount of regional spatial connection to the total area of regional urban areas. As previously described, the most common and important regional spatial connection is economic linkages. Therefore, we use the ratio of total regional economic linkages to the total area of regional urban areas to express the regional spatial compactness. The equation is as follows:(5)$$I=\frac{\sum_{i,j=1}^{n}R_{ij}}{S}$$
where *I* represents the spatial compactness index of the region and *S* represents the total area of the region [26].

To sum up, the Gini coefficient of urban scale based on the dimension of scale distribution reflects the disparity among urban scales in the region, and then is used to measure whether a reasonable urban hierarchy has been formed within the region. The urban primacy index based on the central structure dimension reflects the degree of aggregation of regional comprehensive development elements in the central city, and then is used to measure the radiation and influence of the central city in the region. The index of regional economic linkage strength based on spatial connection dimension reflects the circulation of various factors of production among the cities in the region, and then is used to measure the degree of interconnection among the cities in the region. The spatial compactness index based on spatial compactness dimension reflects the spatial concentration of development factors such as towns, enterprises, factories, funds, facilities, technology and talents in the region, and then is used to measure the spatial compactness of the region.

### 3.2. Comprehensive Evaluation of Sustainable Development

#### 3.2.1. Super-Slack Based Measure-Undesirable Evaluation Model

Data envelopment analysis (DEA) is an effective, non-parametric method to solve comprehensive evaluation problems characterized by multi-input, multi-output, and complex systems. Based on relative efficiency, a mathematical programming model is used to evaluate a decision-making unit (DMU) with multiple inputs and outputs from the perspective of inputs and outputs. DEA can eliminate the influence arising from the dimensions of both subjective and objective factors. While measuring the relative efficiency of each DMU, DEA can identify reasons and magnitudes of inefficiency, in addition to providing a basis for improving DMU efficiency. Insofar of these advantages, DEA requires that the product of the number of input and output indicators be less than the number of DMUs, and that the number of input and output indicators be less than twice the total number of DMUs. That is, $\max\left\{m\times q,2\times \left(m+q\right)\right\}<n$ [27]. If the number of evaluation indicators is too large, most or even all DMUs will be effective. The construction of a comprehensive and objective evaluation index system for regional sustainable development often includes a number of input and output indicators. Therefore, on the basis of the established evaluation index body, this paper adopts the method of combining subjective and objective weights to generate a comprehensive index of DEA input and output. Thus, it provides ideas for the application of the DEA method in the comprehensive evaluation of sustainable development. Based on Tone’s SBM-Undesirable model and the basic idea of super-efficiency evaluation model established by Andersen and Petersen, this paper introduces the Super-SBM-Undesirable evaluation model [28]:
(6)$$\min\rho =\frac{1-\frac{1}{m}\sum_{i=1}^{m}\frac{s_{i}^{-}}{x_{ik}}}{1+\frac{1}{q_{1}+q_{2}}(\sum_{r=1}^{q_{1}}\frac{s_{r}^{g+}}{y_{rk}}+\sum_{t=1}^{q_{2}}\frac{s_{t}^{b-}}{y_{tk}})}\;s.t.\left\{\begin{array}{c} X\lambda +s^{-}=x_{k}\\ Y^{g}\lambda -s^{g+}=y_{k}^{g}\\ Y^{b}\lambda +s^{b-}=y_{k}^{b}\\ s^{-},s^{g+},s^{b-},\lambda \geq 0 \end{array}\right.$$
where ρ is the comprehensive technical efficiency value of the target DMU, λ is the weight vector, *k* is the evaluation unit, *x*, *y^g^* and *y^b^* are the input value, expected output value and non-expected output value respectively; $s^{-}$, $s^{g+}$, $s^{b-}$ are the relaxation of input, expected output and non-expected output respectively. It can be seen from Equation (6) that the SBM-Undesirable model directly puts the relaxation of input and output into the objective function, measures the gap between relaxation variables and optimal production frontier, and solves the relaxation problem of input and output in the traditional DEA model. Simultaneously, it also solves the problem of comprehensive technical efficiency evaluation under unexpected outputs. The core idea of the super efficiency DEA evaluation model is to exclude the evaluated DMU from the reference set, that is, the efficiency of DMU is derived from the frontier of other DMUs [29]. Based on the above principles, the Super-SBM-Undesirable evaluation model derived from SBM-Undesirable evaluation model and super-efficiency evaluation model is defined as follows:(7)$${\min\rho }^{*}=\frac{1+\frac{1}{m}\sum_{i=1}^{m}\frac{s_{i}^{-}}{x_{ik}}}{1-\frac{1}{q_{1}+q_{2}}(\sum_{r=1}^{q_{1}}\frac{s_{r}^{g+}}{y_{rk}}+\sum_{t=1}^{q_{2}}\frac{s_{t}^{b-}}{y_{tk}})}\;s.t.\left\{\begin{array}{c} \overset{n}{\underset{j=1,j\neq k}{\sum }}x_{ij}\lambda_{j}-s_{i}^{-}\leq x_{ik}\\ \overset{n}{\underset{j=1,j\neq k}{\sum }}y_{rj}^{g}\lambda_{j}+s_{r}^{g+}\geq y_{rk}^{g}\\ \overset{n}{\underset{j=1,j\neq k}{\sum }}y_{tj}^{b}\lambda_{j}-s_{t}^{b-}\leq y_{tk}^{b}\\ 1-\frac{1}{q_{1}+q_{2}}(\overset{q_{1}}{\underset{r=1}{\sum }}\frac{s_{r}^{g+}}{y_{rk}}+\overset{q_{2}}{\underset{t=1}{\sum }}\frac{s_{t}^{b-}}{y_{tk}})>0\\ s^{-},s^{g+},s^{b-},\lambda \geq 0\\ i=1,2,\ldots ,m;r=1,2,\ldots ,q;j=1,2,\ldots n\left(j\neq k\right) \end{array}\right.$$

The variable meanings of the above model are consistent with Formula 6.

#### 3.2.2. Evaluation Index System

Based on International Urban Sustainability Indicators List (IUSIL) and the 2011 McKinsey’s Sustainable Development Index in addition to other relevant literature, and following the three principles of target system design (purpose, comparability, and operability), this paper establishes a comprehensive evaluation index system for sustainable development of the YREB, which includes input(X), expected output(Y), and non-expected output(Y) [30], as shown in Table 1. The aim is to systematically and scientifically reflect the sustainable development of the YREB, with input and output as the main line and resource consumption, urban construction, economic development, and pollution disasters as the basic elements. In order to ensure the scientific establishment of the index system, the final determination of the index system also solicits the opinions of many experts from universities, scientific research institutes, planning departments, environmental departments, statistical departments, and so on.

### 3.3. Grey Relational Model

While the most common correlation analysis methods in mathematical statistics mainly include correlation, factor, and principal component analysis, the use of the above methods, however, requires a large amount of relevant data as a prerequisite. Therefore, the grey relational analysis method to overcome this shortcoming has been applied more and more in cases of “small sample, poor information”. Grey relational analysis is a comparative factor analysis method which aims to quantitatively describe the relative changes between the observed sequences in the development of the system to determine whether there is a close relationship between the sequences. Through a comparative analysis, the degree of the relationship between the impact and the target is ranked. If the two observation sequences have similar change characteristics, it shows that they have a greater degree of correlation, and vice versa. The main result of this model is to determine the primary–secondary, superior–inferior relationship among the various influencing factors [31].

Based on different research perspectives, the measurement results of the grey relational model can be divided into a narrow relational degree and a generalized relational degree [32]. The generalized correlation degree includes grey absolute, relative, and comprehensive correlation degrees. Among them, the grey comprehensive correlation degree can not only reflect the similarity degree between the two observation sequences, but also reflect the similarity degree of their change trend [33]. Therefore, this paper chooses the grey comprehensive correlation degree to measure the relationship between the four spatial structure indicators of the YREB and its sustainable development. In combination with the existing research results, the calculation process of grey comprehensive correlation degree is as follows [34]:

The grey absolute relational degree is calculated by constructing a grey relational model. The grey absolute correlation degree measures the similarity between the four spatial structure indices and the sustainable development level in the grey correlation model.

Assuming that the length of $X_{j}$ and $X_{i}$ is the same and the sequence is the sum of the same moments, $X_{j}^{0}$ = ($X_{j}^{0}\left(1\right)$, $X_{j}^{0}\left(2\right),\cdot \cdot \cdot X_{j}^{0}\left(n\right)$), $X_{i}^{0}$ = ($X_{i}^{0}\left(1\right)$, $X_{i}^{0}\left(2\right),\cdot \cdot \cdot X_{i}^{0}\left(n\right))$ are zero portraits of the starting points of $X_{j}$ and $X_{i}$ respectively, and the absolute grey correlation degree between $X_{j}$ and $X_{i}$ is denoted as $\varepsilon_{ji}$:(8)$$\varepsilon_{ji}=\frac{1+\left|S_{j}\right|+\left|S_{i}\right|}{1+\left|S_{j}\right|+\left|S_{i}\right|+\left|S_{i}-S_{j}\right|}$$
(9)$$\left|S_{j}\right|=\left|\overset{n-1}{\underset{k=2}{\sum }}X_{j}^{0}(k)+\frac{1}{2}X_{j}^{0}\left(n\right)\right|$$
(10)$$\left|S_{i}\right|=\left|\overset{n-1}{\underset{k=2}{\sum }}X_{i}^{0}(k)+\frac{1}{2}X_{i}^{0}\left(n\right)\right|$$
(11)$$\left|S_{i}-S_{j}\right|=\left|\overset{n-1}{\underset{k=2}{\sum }}(X_{i}^{0}(k)-X_{j}^{0}(k))+\frac{1}{2}(X_{i}^{0}\left(n\right)-X_{j}^{0}\left(n\right)\right|$$
where $S_{i}$ represents the first observation value of spatial structure sequence i, and there are n observations. The greater the absolute grey correlation degree is, the more similar the spatial structure index is to the level of sustainable development, the closer the relationship between them is, and vice versa.

The grey relational model is constructed to calculate the relative grey relational degree. The grey relative correlation degree calculates the correlation degree of the change rate between the four spatial structure indexes and the level of sustainable development in the grey correlation model. $X_{j}^{\prime }$ and $X_{i}^{\prime }$ are the initial images of $X_{j}$ and $X_{i}$, respectively, and the absolute grey correlation degree of $X_{j}^{\prime }$ and $X_{i}^{\prime }$ is the relative grey correlation degree of $X_{j}$ and $X_{i}$, which is recorded as $r_{ji}$.
(12)$$r_{ji}=\frac{1+\left|S_{j}^{\prime }\right|+\left|S_{i}^{\prime }\right|}{1+\left|S_{j}^{\prime }\right|+\left|S_{i}^{\prime }\right|+\left|S_{i}^{\prime }-S_{j}^{\prime }\right|}$$
(13)$$\left|S_{j}^{\prime }\right|=\left|\overset{n-1}{\underset{k=2}{\sum }}X_{j}^{\prime 0}(k)+\frac{1}{2}X_{j}^{\prime 0}\left(n\right)\right|$$
(14)$$\left|S_{i}^{\prime }\right|=\left|\overset{n-1}{\underset{k=2}{\sum }}X_{i}^{\prime 0}(k)+\frac{1}{2}X_{i}^{\prime 0}\left(n\right)\right|$$
(15)$$\left|S_{i}^{\prime }-S_{j}^{\prime }\right|=\left|\overset{n-1}{\underset{k=2}{\sum }}(X_{i}^{\prime 0}(k)-X_{j}^{\prime 0}(k))+\frac{1}{2}(X_{i}^{\prime 0}\left(n\right)-X_{j}^{\prime 0}\left(n\right)\right|$$
where the calculation of $S_{j}^{\prime }$ and $S_{i}^{\prime }$ is similar to that of $S_{j}$ and $S_{i}$ in the calculation of grey absolute correlation measure. Only $X_{j}$ and $X_{i}$ become the initial values of spatial structure index and sustainable development level, such as $X_{j}^{\prime }$, $X_{i}^{\prime }$. The bigger the grey relative correlation degree is, the more similar the change rate between the spatial structure index and the sustainable development level is, the closer the relationship is, and vice versa.

According to the degrees of both absolute and relative grey correlations obtained from the first step and the second step, the comprehensive grey correlation degree can be calculated. The grey comprehensive correlation degree is the calculation of the weighted average of grey absolute correlation degree and grey relative correlation degree, which can reflect the similarity degree between sequences. At the same time, it also reflects the proximity of the sequence to the change rate of the starting point and can more comprehensively represent the degree of close association between the sequences. The calculation formula is as follows:(16)$$\rho_{ji}=\theta \varepsilon_{ji}+\left(1-\theta \right)r_{ji}$$
where θ is the weight of the absolute grey correlation degree, which is 0.5 in most cases. It means that the absolute correlation degree and the relative correlation degree have the same emphasis. The bigger the grey comprehensive correlation degree is, the closer the relationship between spatial structure index and sustainable development level is, and vice versa.

### 3.4. Data Description

Statistical data is from the nine provinces and two cities of the YREB (where in particular, the spatial structure of Chongqing and Shanghai is based on their respective administrative regions or counties) over the 2006–2015 period. Because some cities had some adjustment to their administrative periods during this period, to ensure the unity of the study, this paper carries out the following treatment: Bijie and Tongren areas in Guizhou Province were changed to prefecture-level cities in 2011 and 2012, respectively. Before that period, there were no statistics available on Bijie and Tongren, so they were eliminated. As a result, the study area concerns 108 urban areas at or above the prefecture level. Finally, the study area involves 108 urban units at or above the prefecture level. In 2011, the Chinese government abolished Chaohu City, Anhui Province, and classified Lujiang County, Wuwei County, Hanshan County, and He County under their original jurisdictions as Hefei City, Wuhu City, and Ma’anshan City, respectively. Therefore, based on the administrative division of Anhui Province in 2011, this paper adjusts the division scope of Hefei City, Wuhu City, and Ma’anshan City. Data from 108 cities at or above the prefecture level are directly extracted from the 2007–2016 China Urban Statistics, Energy Statistics, Environmental Statistics yearbooks, in addition to the relevant statistical yearbooks of the 11 provinces and municipalities. Some of these cities may have data gaps and these are filled by using linear interpolation.

## 4. Empirical Analysis

According to the determined spatial structure index and the obtained statistical data, this paper first estimates the spatial structure of the 11 provinces and cities in the YREB according to Equations (1)–(5). Based on the comprehensive evaluation index system of regional sustainable development and combined with the derived Super-SBM-Undesirable evaluation model, the sustainable development status of the 11 provinces and cities in the YREB is evaluated. Based on the grey correlation model, the sustainable development level of each province and city is taken as the reference sequence, and the spatial structure index of four major regions of each province and city is taken as the comparison sequence, and the correlation degree of each comparison sequence and reference sequence is calculated by the incremental window length method. Four spatial structure indicators are ranked according to correlation degrees, and the relationship between the spatial structure and sustainable development of the YREB is quantitatively analyzed. The influence of the four spatial structure indicators of the YREB on regional sustainable development is discussed in depth.

### 4.1. Overall Analysis

As mentioned above, results are displayed below in Table 2 (due to limitations of space, the results of the spatial structure and social and economic development efficiency of the 11 provinces and cities are not listed):

According to the grey comprehensive correlation degree of sustainable development level and four spatial structures in different periods of the YREB listed in Table 2, it shows that the relationship between the spatial structure and sustainable development level of the YREB is relatively stable in five different stages. From largest to smallest, ranking results are the Gini coefficient of urban scale, urban primacy, regional economic linkage strength, and spatial compactness.

Results show that based on equilibrium dimension, the Gini coefficient of urban scale is the most important spatial structural factor affecting YREB sustainable development. That is, in an ideal, healthy area, cities are neither too tightly packed nor too dispersed. Only when reasonable hierarchies are formed and the various factors of production and the division of labor is evenly distributed throughout the different cities in a given region can we improve overall output levels, narrow inter-city gaps and realize sustainable development of a region. From another point of view, based on the perspective of regional structure, in order to promote YREB sustainable development and ensure the continuity of a clear river, it is necessary to strengthen core cities, improve large and medium-sized cities, further develop small and medium-sized cities, and foster the growth of backward areas. Simultaneously, we should build a clear, scientific, and rational urban hierarchy system, strengthen the regional division of labor and cooperation, promote the optimal allocation of resources, and realize the balanced development of the YREB.

Second only to the Gini coefficient of urban scale, urban primacy based on the central structural development is the spatial structural factor which affects YREB sustainable development. According to the growth pole theory, regional economic growth diffuses outward from nodes, promotes the circulation and diffusion of production factors, which then drives regional development. In the process of regional development, the agglomeration of various factors of production to the first city can produce economies of agglomeration and scale. On the one hand, it can save transportation costs and minimize energy consumption, effectively utilize and allocate limited resources, and form a more effective market and closer industrial correlation effect. On the other hand, it is also conducive to the spillover of knowledge and technology and the improvement of the efficiency of infrastructure and land use. However, if it agglomerates excessively, it will bring about “urban diseases”, such as traffic congestion, housing shortage, and scarcity of resources, which will offset the economic benefits of various factors of production agglomeration to a certain extent. Therefore, cultivating regional growth poles and promoting regional spatial links and diffusion in the YREB are the key steps in promoting regional sustainable development.

The spatial compactness index based on the regional connection dimension and spatial compactness index based on the spatial compactness dimension are also important spatial structural factors affecting the sustainable development of the YREB. The interconnection between cities is the internal driving force for the emergence and development of the external form of the regional spatial structure. With the rapid development of economies, the links between cities in the YREB are getting closer and closer. Through convection, radiation, and transmission, the vertical and horizontal links among cities form a complex network. The effects of polarization and radiation, division of labor and cooperation, and system enhancement can be obtained through the agglomeration of various factors of production in the region and the flow between cities, thus enhancing the overall efficiency of the whole region. As compared with the decentralized areas, more compact areas are resource-saving, high-density, and high-efficiency facets of urban architecture. The compactness of space is the basis of promoting the intensive and efficient development of the region. A moderately compact urban system based on the premise of curbing urban expansion can centralize public facilities, effectively reduce traffic distance and pollution emissions, thus promoting the efficient use of resources, high spatial output, and high-density development. However, an excessively compact space can also lead to uneconomical agglomeration, such as traffic congestion, excessive concentration of industry and population. Therefore, in the next stage, the YREB needs to plan a scientific transportation network, optimize the spatial diffusion axis, build a transportation system with distinct characteristics of “high efficiency, green and environmental protection”, and maximize the reduction of operating costs, so as to strengthen the links between the cities and towns in the YREB and maximize economic benefits. At the same time, in order to promote the sustainable development of the YREB, we must adhere to the idea of high-quality urbanization, curb the unrestrained expansion of cities, adhere to the concept of intensive development, realize land, space, population, and industrial intensive development, and realize the new urbanization development path of intensive use of space, industrial intensive development, and resource intensive recycling.

### 4.2. Separate Analysis

As shown in Table 3, according to the results of the grey comprehensive correlation measure of sustainable development level and spatial structure of the 11 provinces and municipalities in the YREB, this paper classifies the relationship between spatial structure and sustainable development into the following four categories: (1) Gini coefficient of urban scale > urban primacy > regional economic linkage strength > spatial compactness. (2) Urban primacy > Gini coefficient of urban scale > regional economic linkage strength > spatial compactness. (3) Regional economic linkage strength > spatial compactness > Gini coefficient of urban scale > urban primacy. (4) Spatial compactness > regional economic linkage strength > urban primacy > Gini coefficient of urban scale. Among them, the highest frequency is Gini coefficient of urban scale > urban primacy > regional economic linkage strength > spatial compactness, which totally occurs 36 times. There were 14 occurrences of urban primacy > Gini coefficient of urban scale > regional economic linkage strength > spatial compactness. The ranking results are: Regional economic linkage strength > spatial compactness > Gini coefficient of urban scale > urban primacy, appearing four times. The final ranking results are: Spatial compactness > regional economic linkage strength > urban primacy > Gini coefficient of urban scale appearing once.

On the other hand, the ranking results of the 11 provinces and cities in the YREB are relatively stable. They show that in the past development process, the relationship between the spatial structure of provinces and cities and regional sustainable development has remained relatively stable. Of the 55 ranking results, only five changed successively. They are: The period of 2006–2013 in Guizhou Province, the period of 2006–2015 in Sichuan Province, the period of 2006–2015 in Jiangxi Province, the period of 2006–2014 in Shanghai, and the period of 2006–2012 in Yunnan Province. Among them, the period of 2006–2013 in Guizhou and 2006–2015 in Sichuan Province changed from urban primacy > Gini coefficient of urban scale > regional economic linkage strength > spatial compactness to Gini coefficient of urban scale > urban primacy > regional economic linkage strength > spatial compactness. The Gini coefficient of urban scale has replaced the urban primacy as the most closely related spatial structure affecting the sustainable development of Guizhou Province and Sichuan Province. The focus of work has shifted to the construction of a reasonable urban hierarchy system. During the period of 2006–2015 in Jiangxi Province and 2006–2014 in Shanghai, the Gini coefficient of urban scale > urban primacy > regional economic linkage strength > spatial compactness changed to urban primacy > Gini coefficient of urban scale > regional economic linkage strength > spatial compactness. The most closely related factors of spatial structure change from the Gini coefficient of urban scale to urban primacy, and the focus of work shifts to fostering regional growth poles. During the period of 2006–2012, Yunnan Province changed from spatial compactness > regional economic linkage strength > urban primacy > Gini coefficient of urban scale to regional economic linkage strength > spatial compactness > Gini coefficient of urban scale > urban primacy. The regional economic linkage strength has become the most important factor affecting the spatial structure of sustainable development in Yunnan Province. The focus of work is to plan a scientific transportation network and strengthen regional linkages. Finally, in the past development process of Anhui, Hubei, Hunan, Jiangsu, Zhejiang, and Chongqing, the impact of spatial structure on regional sustainable development has not changed. From strength to weakness, the correlation is: Gini coefficient of urban scale > urban primacy > regional economic linkage strength > spatial compactness. In the next stage, in order to accelerate the sustainable development of these six provinces and cities, we should attach great importance to the balanced development of the region, rationally allocate various factors of production, and form a reasonable hierarchy and division of labor system in the region.

## 5. Conclusions

A scientific and reasonable spatial structure can effectively improve the efficiency of regional development and is the “regulator” and “booster” to promote regional sustainable development. Based on the analysis of the relationship between spatial structure and sustainable development of the Yangtze River Economic Belt, this paper puts forward four policy suggestions, namely, building a reasonable urban system, cultivating regional growth poles, strengthening regional linkages and promoting the high-quality development of regional urbanization, respectively, based on the four spatial structure indicators of Gini coefficient of urban scale, urban primacy, regional economic linkage strength and spatial compactness. In this way, we can rationally optimize and regulate the spatial structure of the YREB and by making the spatial interaction reach the best state, we can maximize the allocation of regional resources and spatial synergies, and accelerate the sustainable development of the YREB.

### 5.1. Constructing a Reasonable Urban System

Through the analysis of the relationship between the spatial structure and sustainable development of the YREB, it is found that on the one hand, the Gini coefficient of urban scale is the highest among the four spatial structure indicators and the overall sustainable development of the YREB. Based on this, the key step to promote the sustainable development of the YREB is to construct a reasonable urban hierarchy, strengthen the regional division of labor and cooperation, and realize the optimal allocation of resources. On the other hand, the Gini coefficient of urban scale is also the closest spatial structure factor affecting the sustainable development of Anhui, Guizhou, Hubei, Hunan, Jiangsu, Sichuan, and Chongqing. Therefore, based on the regional spatial structure to promote the sustainable development of these seven provinces and cities, it is also necessary to build a reasonable urban hierarchy system. To sum up, to construct a reasonable urban system in the YREB, firstly, when improving the urban system, the YREB should adhere to the principle of “expanding core cities, improving large and medium-sized cities, developing small and medium-sized cities, cultivating growth poles in backward areas, forming a scientific and rational urban hierarchy system with distinct levels”, and adhere to the planning pattern of “one axis, two wings, three poles and multiple points”. Secondly, the should strengthen the support to the four western provinces and cities. The government should allow the eastern and central regions to promote the development of the western region and actively guide the transfer of overcapacity industries from eastern to western regions. In view of the underdeveloped cities in the western region, the Chinese government should give some support in policy, capital, and technology, so as to narrow the gap between the 11 provinces and cities in the YREB and promote the optimization and regulation of the urban hierarchy system in the YREB through balanced development. Thirdly, a central government-led coordination mechanism should provide the necessary provincial linkages to allow isolated administrative departments and functional units to be amalgamated into a unified network, leading to the breakdown of administrative barriers, reductions in the cost of approval and improvements in operational efficiency. Fourthly, according to the current development situation of the YREB, the Chinese government should properly strengthen the investment of various production factors in some backward cities, promote the complementarity of advantages among cities by adjusting industrial structure, clarifying urban functions and highlighting urban advantages, enhance core competitiveness, and promote the balanced development of the region.

### 5.2. Fostering Regional Growth Poles

Through the above correlation analysis, it was found overall, urban primacy and sustainable development in the YREB are highly correlated, where, for example, it is the closest spatial structure factor that affects the sustainable development of Jiangxi, Shanghai, and Zhejiang. According to the growth pole theory, regional economic growth diffuses from external nodes, driving the internal development of nearby cities. Therefore, cultivating regional growth poles is a key step in promoting the links and diffusion of various production factors. To cultivate the growth poles in the YREB, we can start from the following three points: First, adhere to the “one axis, two wings, three poles, multi-point” planning pattern. At the same time, relying on the Yangtze River Gold Waterway, all kinds of production factors incline to Shanghai, Wuhan, and Chongqing, so as to enhance the core competitiveness of the three major cities, actively participate in international cooperation and competition, and make them become a national and even international metropolis. And pay attention to the radiation effect of the three big cities, so as to promote the sustainable development of the YREB. Secondly, we should strengthen the core competitiveness of Nanjing, Hangzhou, Changsha, Nanchang, Kunming, Chengdu and other sub-core cities in the YREB. The infrastructure of these cities should be strengthened, their functions clearly identified and then improve the role radiation plays in secondary core cities to the surrounding small and medium-sized cities.

### 5.3. Enhancing Regional Spatial Linkages

According to the previous analysis of the relationship between the spatial structure and sustainable development, it is found that the strength of economic linkages is also highly correlated with sustainable development. Therefore, to promote sustainable development in the YREB, it is important to optimize transportation networks and construct them to be highly efficient and environmentally friendly, in addition to strengthening exchanges between cities and towns. It was also identified that the intensity of spatial connection is the spatial structural factor affecting sustainable development in Yunnan Province. Therefore, from the perspective of spatial structure, improving transportation and strengthening infrastructure and spatial connections among cities in the region all help to promote sustainable development.

Optimizing and perfecting the transportation system of the YREB and strengthening regional spatial relations would mainly rely on water and railway transport lines. Firstly, from the perspective of waterway transportation, the Yangtze River Gold Waterway runs across China’s east and west, linking 11 provinces and cities, carrying the important task of circulation and allocation of various factors of production in the YREB. On the basis of a comprehensive and objective analysis of the current situation, in addition to the nature and functions YREB basins, it is suggested that through a rational layout and planning, the dredging of waterways can improve cargo transport volumes. Simultaneously, infrastructure at the Ports of Wuhan, Chongqing, and other important ports can be improved and the construction of an efficient waterway transportation network would lead to higher connectivity between YREB note cities. The accelerated construction of railway transportation, most notably the Shanghai-Chengdu high-speed railway along the Yangtze River should strengthen the links between the eastern and western regions. Similarly, the accelerated construction of the Shanghai-Kunming high-speed railway would strengthen communication between cities south of the Yangtze River. Upgrades to the railway connections between Chengdu-Chongqing, central Yunnan, Chongqing-Kunming and Chongqing-Guizhou should be made to promote the formation of the “1+1+3” urban agglomeration (composed of Kunming, Guiyang and Huaihua, and Chengdu with Chongqing as the core) in southwest China. In addition, we should intensify reform and innovation, build comprehensive transport hubs, strengthen multimodal transport by water, road and air, and build safe, efficient, green and environmentally friendly three-dimensional transport corridors, so as to strengthen the inter-regional links and promote the sustainable development of the YREB.

### 5.4. Promoting the High-quality Development of Urbanization

Spatial compactness is also an important factor to promote the sustainable development of the YREB. A moderately compact urban system can promote efficient utilization of resources, high spatial output, and high-density development. However, at the same time, we need to avoid uneconomical effects, such as traffic congestion and the excessive concentration of industry and population. Therefore, in order to achieve the sustainable development of the YREB, we must adhere to the high-quality development of urbanization, minimize blind urban expansion, adhere to the concept of intensive development, take the new road of urbanization development of land, space, population and industrial intensive development. We should also speed up the transformation of the urbanization development mode in the YREB by deepening reform in an all-round way and by removing institutional obstacles. From the extensive development mode of high investment, high pollution, low benefit, and low level to the development mode of intensive, efficient, quality-oriented, and benefit-oriented coordination, thus realizing the sustainable development of the YREB.

## Figures and Tables

**Table 1 ijerph-16-03076-t001:** Evaluation index system for regional sustainable development.

Type	First Level Index	Second Level Index	Third Level Index
Input	Natural resource consumption (X_1_)	Land (X_11_)	Land area per capita (X_111_)
		Water (X_12_)	Per capita water consumption (X_121_)
		Energy (X_13_)	Total energy consumption per capita (X_131_)
	Social resource consumption (X_2_)	Capital (X_21_)	Investment in environmental pollution control accounts for GDP share (X_211_)
			Investment in fixed assets per capita (X_212_)
		Labor (X_22_)	Employment ratio (X_221_)
Desirable output	Social development (Y_1_)	Urban development (Y_11_)	Urban road area per capita (Y_111_)
			Green coverage area per capita (Y_112_)
			Urban area per capita (Y_113_)
			Urban population ratio (Y_114_)
		Education, technology, culture and health care (Y_12_)	Three kinds of patent authorization per capita in China (Y_121_)
			Number of health workers per 10000 people (Y_122_)
			Consumption of education, culture and entertainment per capita (Y_123_)
			Number of full-time teachers in Colleges and universities per ten thousand people (Y_124_)
		Social Security (Y_13_)	Basic old-age insurance coverage ratio (Y_131_)
			Unemployment insurance coverage ratio (Y_132_)
			Insurance ratio of medical insurance for urban employees (Y_133_)
			Insurance ratio of industrial injury insurance (Y_134_)
			Birth insurance coverage ratio (Y_135_)
		Living Standards (Y_14_)	consumption expenditure per capita (Y_141_)
			Disposable income per capita (Y_142_)
	Economic development (Y_2_)	Economic Growth (Y_21_)	GDP growth rate (Y_211_)
		Economic Structure (Y_22_)	Third industry share (Y_221_)
		Economic Scale (Y_23_)	GDP per capita (Y_231_)
Undesirable output	Pollution, disasters and accidents (Y_3_)	Water Pollution (Y_31_)	Wastewater discharge per capita (Y_311_)
		Air Pollution (Y_32_)	SO_2_ emissions per capita (Y_321_)
			Smoke and dust emissions per capita (Y_322_)
		Natural Disasters (Y_33_)	Direct economic losses natural disasters per capita (Y_331_)
		Traffic Accidents (Y_34_)	Direct economic loss traffic accident per capita (Y_341_)

**Table 2 ijerph-16-03076-t002:** Grey comprehensive relevance degree of sustainable development level and spatial structure in different periods of the Jiangsu Economic Belt.

Year	H1	H2	H3	H4	Ranking Result
2006–2011	0.814	0.835	0.621	0.596	H2 > H1 > H3 > H4
2006–2012	0.776	0.795	0.601	0.574	H2 > H1 > H3 > H4
2006–2013	0.746	0.768	0.584	0.560	H2 > H1 > H3 > H4
2006–2014	0.737	0.757	0.574	0.553	H2 > H1 > H3 > H4
2006–2015	0.724	0.750	0.567	0.549	H2 > H1 > H3 > H4

H1: urban primacy; H2: Gini coefficient of urban scale; H3: regional economic linkage strength; H4: spatial compactness.

**Table 3 ijerph-16-03076-t003:** Grey comprehensive relevance degree of sustainable development level and spatial structure in the 11 provinces and cities of the Yangtze River Economic Belt at different stages.

Region	2006–2011	2006–2012
H1	H2	H3	H4	Ranking Result	H1	H2	H3	H4	Ranking Result
Anhui	0.856	0.859	0.620	0.549	H2 > H1 > H3 > H4	0.836	0.837	0.596	0.538	H2 > H1 > H3 > H4
Guizhou	0.937	0.912	0.600	0.597	H1 > H2 > H3 > H4	0.953	0.950	0.567	0.560	H1 > H2 > H3 > H4
Hubei	0.749	0.782	0.575	0.555	H2 > H1 > H3 > H4	0.692	0.724	0.567	0.549	H2 > H1 > H3 > H4
Hunan	0.860	0.894	0.553	0.536	H2 > H1 > H3 > H4	0.797	0.855	0.532	0.521	H2 > H1 > H3 > H4
Jiangsu	0.770	0.824	0.573	0.565	H2 > H1 > H3 > H4	0.730	0.786	0.553	0.547	H2 > H1 > H3 > H4
Jiangxi	0.826	0.837	0.754	0.656	H2 > H1 > H3 > H4	0.781	0.789	0.702	0.617	H2 > H1 > H3 > H4
Shanghai	0.849	0.930	0.580	0.579	H2 > H1 > H3 > H4	0.822	0.864	0.569	0.569	H2 > H1 > H3 > H4
Sichuan	0.787	0.773	0.651	0.575	H1 > H2 > H3 > H4	0.775	0.738	0.602	0.550	H1 > H2 > H3 > H4
Yunnan	0.652	0.648	0.827	0.855	H4 > H3 > H1 > H2	0.630	0.631	0.831	0.775	H3 > H4 > H2 > H1
Zhejiang	0.800	0.795	0.550	0.544	H1 > H2 > H3 > H4	0.728	0.713	0.549	0.542	H1 > H2 > H3 > H4
Chongqing	0.869	0.929	0.549	0.542	H2 > H1 > H3 > H4	0.790	0.854	0.546	0.540	H2 > H1 > H3 > H4
**Region**	**2006–2013**	**2006–2014**
**H1**	**H2**	**H3**	**H4**	**Ranking Result**	**H1**	**H2**	**H3**	**H4**	**Ranking Result**
Anhui	0.827	0.829	0.593	0.536	H2 > H1 > H3 > H4	0.820	0.826	0.588	0.534	H2 > H1 > H3 > H4
Guizhou	0.857	0.957	0.549	0.539	H2 > H1 > H3 > H4	0.787	0.940	0.536	0.527	H2 > H1 > H3 > H4
Hubei	0.650	0.682	0.561	0.545	H2 > H1 > H3 > H4	0.619	0.652	0.558	0.542	H2 > H1 > H3 > H4
Hunan	0.763	0.822	0.522	0.515	H2 > H1 > H3 > H4	0.781	0.849	0.519	0.513	H2 > H1 > H3 > H4
Jiangsu	0.705	0.762	0.538	0.535	H2 > H1 > H3 > H4	0.688	0.745	0.532	0.529	H2 > H1 > H3 > H4
Jiangxi	0.750	0.754	0.663	0.592	H2 > H1 > H3 > H4	0.726	0.728	0.637	0.576	H2 > H1 > H3 > H4
Shanghai	0.802	0.814	0.563	0.562	H2 > H1 > H3 > H4	0.788	0.744	0.566	0.562	H1 > H2 > H3 > H4
Sichuan	0.815	0.751	0.564	0.531	H1 > H2 > H3 > H4	0.951	0.865	0.533	0.516	H1 > H2 > H3 > H4
Yunnan	0.614	0.619	0.780	0.729	H3 > H4 > H2 > H1	0.600	0.610	0.751	0.703	H3 > H4 > H2 > H1
Zhejiang	0.689	0.667	0.546	0.540	H1 > H2 > H3 > H4	0.661	0.636	0.544	0.538	H1 > H2 > H3 > H4
Chongqing	0.730	0.786	0.545	0.539	H2 > H1 > H3 > H4	0.688	0.734	0.545	0.539	H2 > H1 > H3 > H4
**Region**	**2006–2015**					
**H1**	**H2**	**H3**	**H4**	**Ranking Result**					
Anhui	0.815	0.827	0.583	0.532	H2 > H1 > H3 > H4					
Guizhou	0.787	0.919	0.533	0.524	H2 > H1 > H3 > H4					
Hubei	0.597	0.633	0.554	0.540	H2 > H1 > H3 > H4					
Hunan	0.811	0.857	0.518	0.512	H2 > H1 > H3 > H4					
Jiangsu	0.678	0.734	0.528	0.526	H2 > H1 > H3 > H4					
Jiangxi	0.703	0.701	0.622	0.567	H1 > H2 > H3 > H4					
Shanghai	0.788	0.694	0.566	0.566	H1 > H2 > H3 > H4					
Sichuan	0.891	0.972	0.520	0.509	H2 > H1 > H3 > H4					
Yunnan	0.588	0.603	0.729	0.684	H3 > H4 > H2 > H1					
Zhejiang	0.644	0.612	0.542	0.537	H1 > H2 > H3 > H4					
Chongqing	0.660	0.696	0.544	0.538	H2 > H1 > H3 > H4

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
