# Peer review of "Relevance Analysis of Sustainable Development of China’s Yangtze River Economic Belt Based on Spatial Structure"

_ijerph, 2019, doi:10.3390/ijerph16173076_

Round 1

Reviewer 1 Report

Minor comments:

line 133: please, add reference to Zhou

in formula (4) please clarify what are V_i and V_j - the same as G_i and G_j?

Why have signs changed in the objective function (7) relatively to (6)? Is there a typo?

Lines 264 and 278: please introduce the reference

Line 278: should be X_j and epsilon_ji instead of X_0 and epsilon_0i

Line 309: please, use the same symbol for epsilon in (16) and (8)

Author Response

Dear reviewer, thank you very much for your valuable comments and suggestions. According to your proposal, we have made a substantial revision of the paper, so as to get your approval.

Point 1: Line 133: please, add reference to Zhou in formula (4) please clarify what are Vi and Vj - the same as Gi and Gj?

Response 1: According to your suggestion, we add a reference to Zhou in formula (4). Meanwhile, Vi and Vj, Gi and Gj all represent the GDP of cities in the region, which is duplicated. Vi and Vj were deleted during this revision, and the GDP of cities in the region was represented only by Gi and Gj.

Point 2: Why have signs changed in the objective function (7) relatively to (6)? Is there a typo?

Response 2: Because formula 6 is the SBM-Undesirable evaluation model, while formula 7 is the Super-SBM-Undesirable evaluation model. Formula 7 is derived from Formula 6 combined with super efficiency DEA evaluation model.

Point 3: Lines 264 and 278: please introduce the reference.

Response 3: According to your suggestion, we have supplemented relevant references. Please see Line 286 and 290.

Point 4: Line 278: should be X_j and epsilon_ji instead of X_0 and epsilon_0i.

Response 4: According to your suggestion, X_j and epsilon_ji instead of X_0 and epsilon_0i.

Point 5: Line 309: please, use the same symbol for epsilon in (16) and (8).

Response 3: According to your suggestion, we use the same symbol for epsilon in (16) and (8).

Reviewer 2 Report

This paper uses Gini coefficient of urban scale, urban primacy, regional economic linkage strength and spatial compactness to measure the spatial structure of the Yangtze River Economic Belt, including scale distribution, central structure, spatial connection and compactness dimension. The optimized Super-SBM-Undesirable model is performed to evaluate the sustainable development status of the Yangtze River Economic Belt. The authors also adopt grey relational model to calculate the correlational degree between the spatial structure and sustainable development level. The results show that the relationship between the spatial structure and sustainable development level of the YREB is 357 relatively stable in five different stages, and Gini coefficient of urban scale is the 361 most important spatial structural factor affecting YREB sustainable development.

My comments and suggestions are as follows:

1.      The abstract and citation in the text should follow the format of this journal. The abstract lacks of results. In the text, reference numbers should be placed in square brackets [ ].

2.      In general, it should not appear “we see” in the official journal (e.g., page 11 row 356). I suggest the authors revise the language usage.

3.      In the 4.1 overall analysis section, the empirical results of table 2 should present the statistics in the text and further explain its implications for readers.

4.      Overall, this paper is well organized and its results explain the regional economic development by appropriate research methods.

Author Response

Dear reviewer, thank you very much for your valuable comments and suggestions. According to your proposal, we have made a substantial revision of the paper, so as to get your approval.

Point 1: The abstract and citation in the text should follow the format of this journal. The abstract lacks of results. In the text, reference numbers should be placed in square brackets [ ].

Response 1: According to your suggestion, we have revised the abstracts and citations in the text according to the format of your journal. At the same time, we supplement the research conclusions in the summary. Finally, the reference numbers is modified.

Point 2: In general, it should not appear “we see” in the official journal (e.g., page 11 row 356). I suggest the authors revise the language usage.

Response 2: According to your suggestion, we have perfected the language usage.

Point 3: In the 4.1 overall analysis section, the empirical results of table 2 should present the statistics in the text and further explain its implications for readers.

Response 3: According to your suggestion, we have adjusted the demonstration paragraph of the empirical results and improved its interpretation of its implications.

Point 4: Overall, this paper is well organized and its results explain the regional economic development by appropriate research methods.

Response 4: Thank you very much for your recognition of our research results.

Reviewer 3 Report

Viewpoints presented in the introduction section are not supported by sufficient evidences. The significance of the research is not well demonstrated.

The relationship between “regional development” and “sustainable development” is not clearly stated.

Latest research progress in the related fields needs to be reviewed and evaluated.

The choice of the four indicators seems very casual. What is the relationship among these indicators and the relationship between indicators and sustainable development?

Measures in Table 1 should be clearly defined and the data sources need to be clarified one by one.

It seems not necessary to list out so many formulas about well-known methods.

Are the results of H1-H4 comparable? Why is the ranking so important that almost all results are discussed around it?

Policy recommendations are not closely related to empirical results

There are many mistakes in terminology, grammar, and wording.

Author Response

Dear reviewer, thank you very much for your valuable comments and suggestions. According to your proposal, we have made a substantial revision of the paper, so as to get your approval.

Point 1: Viewpoints presented in the introduction section are not supported by sufficient evidences. The significance of the research is not well demonstrated.

According to your suggestion, we have added evidence to support the points raised in the quotation. And we also provide evidence to support the significance of the research.

Point 2: The relationship between “regional development” and “sustainable development” is not clearly stated.

Response 2: According to your suggestion, we have expressed the relationship between "regional development" and "sustainable development". Please see Line 39-49.

Point 3: Latest research progress in the related fields needs to be reviewed and evaluated.

Response 3: According to your suggestion, we have revised the latest research progress in related fields.

Point 4: The choice of the four indicators seems very casual. What is the relationship among these indicators and the relationship between indicators and sustainable development?

Response 4: According to your suggestion, we explained the choice of four indicators. At the same time, we elaborate on the relationship between the four indicators and the relationship between indicators and sustainable development. Please see Line 106-112 and 208-219.

Point 5: Measures in Table 1 should be clearly defined and the data sources need to be clarified one by one. It seems not necessary to list out so many formulas about well-known methods.

Response 5: According to your suggestion, Table 1 is explained and its data source is clearly stated. Please see Line 337-353.

Point 6: Are the results of H1-H4 comparable? Why is the ranking so important that almost all results are discussed around it?

Response 6: H1-H4 represent the correlation strength between the four spatial structure indicators and regional sustainable development. The purpose of this paper is to find out the spatial structure index which can most affect the sustainable development of the region.

Point 7: Policy recommendations are not closely related to empirical results

Response 7: According to your suggestions, we revised the policy recommendations according to the empirical results. According to the order of correlation intensity between the four spatial structure indicators and regional sustainable development, the corresponding policy recommendations have been put forward in turn.

Point 8: There are many mistakes in terminology, grammar, and wording.

Response 8: According to your suggestion, we will improve the terminology, grammar and wording in the paper.    

Round 2

Reviewer 3 Report

No more comments

Author Response

Thank you very much for your recognition of our research results.